electrical engineering/electromagnetism

space–time metamaterials, translation operator, non-reciprocity, time periodic circuits

**Author for correspondence:**
Sameh Y. Elnaggar
e-mail: samehelnaggar@gmail.com

# Properties of translation operator and the solution of the eigenvalue and boundary value problems of arbitrary space–time periodic metamaterials

## Sameh Y. Elnaggar[1] and Gregory N. Milford[2]

[1]Department of Electrical and Computer Engineering, Royal Military College of Canada, Kingston, Ontario, Canada
[2]School of Engineering and Information Technology, University of New South Wales, Canberra, Australia

SYE, 0000-0002-6527-9861

There is a recent interest in understanding and exploiting the intriguing properties of space–time metamaterials. In the current manuscript, the time periodic circuit theory is exploited to introduce an appropriate translation operator that fully describes arbitrary space–time metamaterials. It is shown that the underlying mathematical machinery is identical to the one used in the analysis of linear time invariant periodic structures, where time and space eigen-decompositions are successively employed. We prove some useful properties the translation operator exhibits. The wave propagation inside the space time periodic metamaterial and the terminal characteristics can be rigorously determined via the expansion in the operators eigenvectors (space–time Bloch waves). Two examples are provided that demonstrate how to apply the framework. In the first, a space time modulated composite right left handed transmission line is studied and results are verified via time domain computations. Furthermore, we apply the theory to explain the non-reciprocal behaviour observed on a nonlinear transmission line manufactured in our lab. Bloch-waves are computed from the extracted circuit parameters. Results predicted using the developed machinery agree with both measurements and time domain analysis. Although the analysis was carried out for electric circuits, the approach is valid for different domains such as acoustic and elastic media.

# 1. Introduction

Recently, there has been a surge of interest in studying the properties of space–time modulated systems that arise from the intrinsic asymmetric interaction of the space–time harmonics [1,2]. Such asymmetry breaks the principle of reciprocity, hence enabling the design of novel non-reciprocal devices such as nonreciprocal antenna [3–5], magnet-less circulators [6–8], one-way beam splitters [9], isolators [10,11] and space–time modulated metasurfaces [12,13]. On top of that, systems that possess time and/or space–time periodic elements may not be constrained by the physical limitations of linear time invariant (LTI) systems. For instance, a time modulated reactance does not necessarily result in a total reflection of an incident wave impinging the structure, hence enabling the accumulation of energy [14]. Additionally, a time switched transmission line was demonstrated to have a broadband matching capability not limited by the Bode–Fano criteria that impose a return loss/bandwidth trade-off [15,16]. Taravati & Caloz [17] provide an excellent review on the developments, applications and methods of analysis of space–time media.

The interest in space–time modulated media dates back to the middle of the last century in the context of understanding the properties of distributed parametric amplifiers [18–23]. The salient properties of such media emerge from Bloch-Floquet theory, which states that the eigensolutions are the sum of infinite space time harmonics. Unlike LTI systems, space–time modulation leads to an asymmetric dispersion relation, where at a given frequency $\omega$, the forward and backward wavenumbers are not necessarily equal [1,2].

In general, space–time modulated systems are built from nonlinear lumped elements. The interaction of the nonlinearity and a strong pump results in a spatio-temporal modulation of one or more system parameters [24,25]. In a conventional treatment, it is generally assumed that the system is distributed such that the modulation wavelength is much longer than the length of the unit cell. Such an assumption permits describing the system via partial differential equations (PDEs). Upon the application of the Bloch–Floquet condition, the PDEs reduce to an infinite system of homogeneous equations, where the eigenmodes can be calculated from its non-trivial solutions. Very recently, we have developed a circuit based framework that extends the theory of linear time periodic (LTP) circuits and systems, developed in [26–28], to space–time structures [29]. The dispersion relation emerges from the application of the Bloch condition to a unit cell that links its input and output time harmonics. Therefore, it is valid for both electrically long and electrically short systems. Furthermore, the framework enables the exploration of various structures such as Composite Right Left Handed (CRLH) transmission lines (TLs) and non-sinusoidal periodic modulation [29]. The governing equations reduce to generalized telegraphist's equations when the unit cell is infinitesimally small.

In the current manuscript, we exploit the circuit approach [29] to develop a translation operator and explore its basic properties. We show that the approach is, in principle, equivalent to how periodic structures are described via the diagonalization of the translation operator, where the eigenmodes are nothing but the Bloch waves. Additionally, for a generic unit cell and an arbitrary periodic modulation, the boundary value problem is solved via the expansion of the solution inside the structure in terms of the eigenmodes.

Section 2 starts with a brief review of how the immittance matrix, a generalization of the immittance circuit parameter in LTI systems, emerges from Bloch–Floquet theorem. We then proceed by showing how elements in cascade combine and how the ABCD parameters change between unit cells. In §3, we focus on the eigenvalue problem that describes the system modal behaviour. The invariance of eigenvalues and eigenvectors resulting from the transformation of the system translation operator is presented. We also show that for a generic space–time circuit, the eigensolutions along the modulation line are equivalent. Furthermore, the section discusses how the LTI and LTP systems are formally equivalent in the sense of eigen-decompositions in time and spatial domains. Section 4 demonstrates how the driven modal solution is expanded in terms of the system eigenmodes. Additionally, expressions of the transmission and reflection coefficients are derived. In §5, two systems are studied. In §5.1, a CRLH TL is fully described using the developed machinery and results are compared to time domain simulation. Dispersion relations, eigenvalues, eigenvectors, waveforms and transmission coefficient are computed for both the right-hand (RH) and left-hand (LH) regimes. Subsection 5.2 applies the framework to a nonlinear RH TL that has been fabricated in our lab. The modulation is achieved via a strong pump and hence, the TL operates in the sonic regime [22]. Dispersion relations, eigenvalues and waveforms are computed and compared to simulation. Furthermore, the scattering parameters are calculated and compared to measurements.

## 2. Translational properties of immittance and ABCD matrices

In LTP circuits, the Bloch Floquet theorem allows the voltage and current harmonics to be related via immittance matrices [26,29]. An immittance matrix is best visualized as a generalization of the concept of immittance, a scalar complex quantity, in LTI systems. Before proceeding with the detailed description, it is worth noting that we represent the $(m, n)$ element in matrix $\mathbf{A}$, using the notation $A_m^n$, i.e. the subscript (superscript) represents the row (column).

Without loss of generality, consider a time modulated capacitance $\tilde{C}(t)$. The eigensolutions of LTP systems are in the form of $p(t)\exp(i\omega t)$, $p(t + T) = p(t)$ [29,30], hence the voltage–current relation can be described compactly by the matrix equation

$$\mathbf{I} = \tilde{\mathbf{Y}}\mathbf{V}. \tag{2.1}$$

The entries of an arbitrary $k$th row can be determined from the zeroth row, where

$$\tilde{Y}_k^{k+l}(\omega) = \tilde{Y}_0^l(\omega_k), \tag{2.2}$$

where $\omega_k \triangleq \omega + k\hat{\omega}$.

Consider now a structure where the above capacitance is modulated via a travelling wave with speed $\hat{v}$, i.e.

$$\tilde{C}(t, x) = \tilde{C}\left(t - \frac{x}{\hat{v}}\right),$$

and $x$ is a multiple of the underlying spatial lattice distance $p$ (i.e. $x = np$, $n = -\infty, \ldots, -2, -1, 0, 1, 2, \ldots, \infty$). Again, expanding $\tilde{C}$ in its Fourier components, and noting that the modulation frequency $\hat{\omega}$ and wavenumber $\hat{\beta}$ are related by $\hat{\omega} = \hat{v}\hat{\beta}$, imply that

$$\tilde{Y}_q^p(x) = \tilde{Y}_q^p(0)\, e^{-i[q-p]\hat{\beta}x}. \tag{2.3}$$

This means that the elements in a given row of an admittance matrix $\tilde{\mathbf{Y}}(x)$ are those in the same row of $\tilde{\mathbf{Y}}(0)$, but multiplied by a phasor that rotates in the counter clockwise direction as we go from left to right. Additionally for a fixed column, the elements from top to bottom are multiplied by a clockwise rotating phasor.

In the subsequent development, the ABCD parameters of the unit cell will be shown to play a significant role. They are formed via the multiplication of LTP impedance and admittance matrices. Therefore, it is crucial to understand the properties of the product of matrices representing space–time modulated elements. Consider for instance the series $\mathbf{Z}$ and shunt $\mathbf{Y}$ of a lumped right-handed transmission line. It can be shown that the $(r, r - s)$ element of $(\mathbf{ZY})_r^{r-s}$ at position $x$

$$(\mathbf{ZY})_r^{r-s}(x) = (\mathbf{ZY})_r^{r-s}(0)\, e^{-is\hat{\beta}x}.$$

Therefore, we have the following important property:

**Property 2.1.** For a space–time periodic structure consisting of a cascade of space–time periodic unit cells, the ABCD parameters $\mathbf{X} = \mathbf{A}$, $\mathbf{B}$, $\mathbf{C}$ and $\mathbf{D}$ for a unit cell $x$ away from the origin are related to the ones at the origin by

$$\mathbf{X}_q^p(x) = \mathbf{X}_q^p(0)\, e^{-i[q-p]\hat{\beta}x} = \mathbf{X}_q^p(0)\, \Gamma_{q-p}^{x/p}, \tag{2.4}$$

where $\Gamma_{q-p} \triangleq \exp(-i[q - p]\hat{\beta}p)$. Hence, the ABCD parameters follow the same transformation of immittance matrices (equation (2.3)). Note that if the modulation wavelength $\hat{\lambda} \triangleq 2\pi/\hat{\beta}$ is a multiple of $p$, and when $x$ is a multiple of $\hat{\lambda}$, $\Gamma$ becomes the identity matrix and $\mathbf{X}_q^p(x) = \mathbf{X}_q^p(0)$ as expected.

## 3. Eigenvalue problem and dispersion relation

The harmonics at the terminals of the $n$th unit cell are related by the ABCD transfer matrix

$$\begin{pmatrix} \mathcal{V}[n] \\ \mathcal{I}[n] \end{pmatrix} = \begin{pmatrix} \mathbf{A} & \mathbf{B} \\ \mathbf{C} & \mathbf{D} \end{pmatrix}_n \begin{pmatrix} \mathcal{V}[n+1] \\ \mathcal{I}[n+1] \end{pmatrix},$$

or in the more convenient form

$$\Psi_n = \mathbf{P}_n \Psi_{n+1}, \tag{3.1}$$

where $\Psi_r \triangleq (\mathcal{V}[r], \mathcal{I}[r])^t$ is an infinite-dimensional vector that stores the amplitude of all time harmonics at $x = rp$; and $\mathbf{P}_n$ is the ABCD matrix of the $n$th unit cell.

We seek solutions of the form

$$\Psi_{n+1} = e^{-i\beta p} \Lambda \Psi_n, \tag{3.2}$$

where

$$\Lambda = \begin{pmatrix} \Gamma & 0 \\ 0 & \Gamma, \end{pmatrix}$$

and $\Gamma_{rr} = \Gamma_r = \exp(-ir\hat{\beta}p)$ and zero otherwise. The condition (3.2) is equivalent to seeking a travelling wave solution of the form $\sum_{r=-\infty}^{\infty} \Psi_{0r}\, e^{i[\omega_r t - \beta_r np]}$, where $\beta_r \triangleq \beta + r\hat{\beta}$. Therefore, (3.1) and (3.2) can be combined to give the eigenvalue problem (EVP)

$$\mathbf{T}_n \Psi_n = \mathbf{P}_n \Lambda \Psi_n = e^{i\beta p} \Psi_n. \tag{3.3}$$

Note that $\mathbf{T}_n \triangleq \mathbf{P}_n \Lambda$ acts as a translation operator, where $\exp(i\beta p)$ and $\Psi_n$ are its eigenvalue and eigenvector, respectively. Furthermore, $\mathbf{T}_n$ is a function of the operating frequency $\omega$. If $\mathbf{T}_n$ is indeed a translation operator, we should expect that the eigen-solutions are independent of the unit cell. The next property shows that $\mathbf{T}_n$ is indeed independent of the unit cell (i.e. independent of $n$).

**Property 3.1.** Consider the EVP (3.3) at which $\beta$ and $\Psi_n$ is a solution. Then

$$\beta' = \beta, \tag{3.4}$$

and the eigenvector

$$\underbrace{V'_k}_{\text{at } n+1} = \underbrace{V_k}_{\text{at } n} \Gamma_k, \ \sim\ \underbrace{I'_k}_{\text{at } n+1} = \underbrace{I_k}_{\text{at } n} \Gamma_k \tag{3.5}$$

is a solution of

$$\mathbf{T}_{n+1} \Psi_{n+1} = e^{i\beta' p} \Psi_{n+1}.$$

The proof of property (3.1) is presented in the electronic supplementary material. Equations (3.4) and (3.5) imply that regardless of the unit cell used, the EVP will always result in a unique propagation constant $\beta$. Additionally, the $k$th component of the eigenvector changes in a way that is equivalent to the phase delay of the $k$th harmonic along a unit cell, which is equal to $k\hat{\beta}p$. Therefore, the solution of the EVP (3.3) is independent of the unit cell.

The above result can be generalized to

**Corollary 3.2.** *The solution of (3.3) using* $\mathbf{T}_{n+q}$ *is*

$$\beta' = \beta$$

*and*

$$V'_k = V_k \Gamma_k^q, \ \sim\sim I'_k = I_k \Gamma_k^q.$$

It is worth digressing here and discussing how the above formulation has the same mathematical foundations used to describe LTI systems. The discussion provides a deeper insight into the *mathematical* structure of the framework and highlights the physical similarities and differences between LTI and LTP systems. We refer to figure 1 that depicts a conceptual diagram of the applied flow. Figure 1*a* illustrates the logical flow one usually applies to describe LTI circuits. The basic relations of a two port circuit are given via differential and algebraic equations, leading to an LTI state space representation of the system. This in turn allows the output parameters to be related to the input ones via the convolution with the $2 \times 2$ impulse response matrix, shown in the left side of the figure by the $A(t)$, $B(t)$, $C(t)$ and $D(t)$ functions. The convolution operation with the impulse response can be easily diagonalized, where the set of complex functions $\exp(i\omega t)$ form the eigen-functions of the convolution operator $\mathbf{P}^*$. Hence, it allows the system to be represented by simple

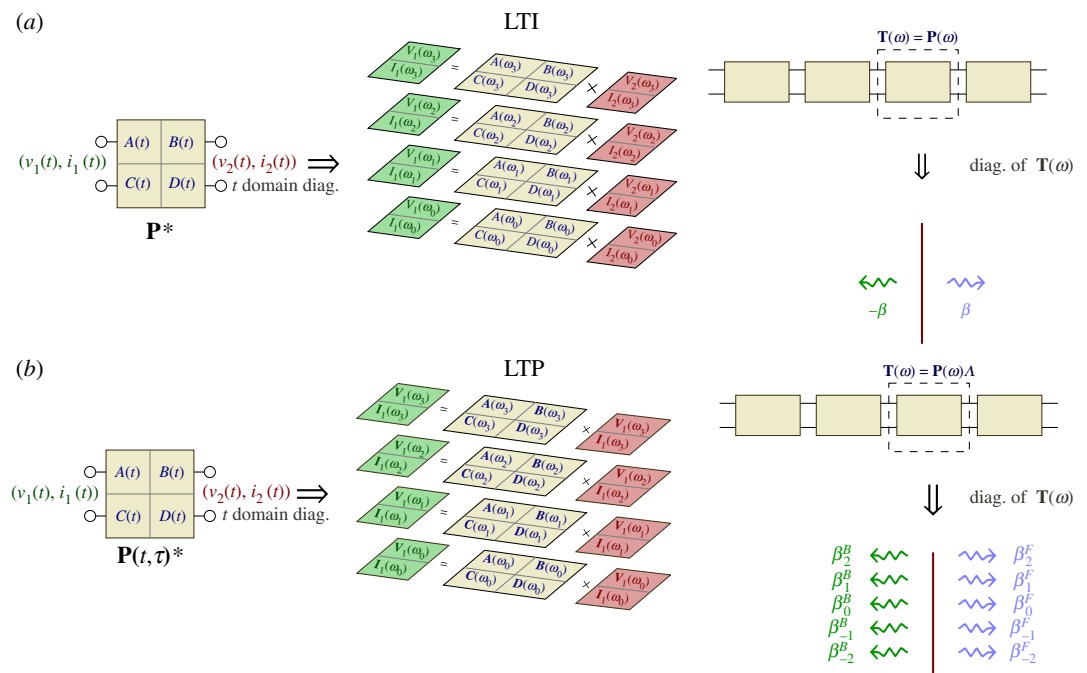

**Figure 1.** Eigen-decomposition of (*a*) LTI and (*b*) LTP systems in the frequency and subsequently in the spatial domains.

matrix multiplications at each frequency $\omega$ as shown in the middle panel of figure 1*a*. In other words, the frequency domain representation is merely an expansion of the input and output parameters in the eigen-functions basis of $\mathbf{P}^*$. For arbitrary time varying circuits, the output and input parameters can be related via the aid of the state transition matrix, rendering the analysis of such general systems a bit challenging. Fortunately for LTP systems, Floquet theorem tells us that the eigen-functions are in the form $\sum Q_r \exp(i\omega_r t)$. Therefore, the input and output voltages and currents become infinitely long complex vectors that store the coefficients $Q_r$ and are connected via the multiplication with LTP ABCD matrix, as equation (3.1) emphasizes. In another words, the LTP ABCD matrix is the operator that connects the output and input vectors expanded in the LTP temporal eigen-functions.

When two port LTI networks are cloned to form a periodic structure, the network ABCD matrix does not change as one moves from one unit cell to the other. Such ABCD matrix maps the input applied to one of its terminals to the output, hence it represents the translation operator $\mathbf{T}$ (i.e. $\mathbf{T} = \mathbf{P}$). $\mathbf{T}$ is a linear operator in the two-dimensional complex space $\mathcal{C}^2$. Diagonalization of $\mathbf{T}$ is equivalent to finding its eigen-functions, which are nothing but the Bloch waves. The eigenvalues are conveniently represented as $\exp(i\beta p)$, where $p$ is the unit cell length. Therefore for each $\omega$, representing a temporal eigenvalue, there are two spatial eigenvalues $\pm\beta$ that diagonalize the spatial operator. The ordered pair $(\omega, \pm\beta)$ describes the propagation of waves in a periodic structure at $\omega$. On the other hand, property 2.1 implies that the ABCD matrix $\mathbf{P}_n$ in a space–time modulated structure changes from one point to the other (i.e. depends on $n$). Nevertheless, property 3.1, along with equation (3.3) shows that $\mathbf{T}_n$ does indeed act as a translation operator. Since the space of time harmonics is infinite, the diagonalization of $\mathbf{T}_n$ (i.e. enforcing the Bloch condition) results in an infinite number of eigen-functions as the last panel of figure 1*b* illustrates. Note that if the modulation is removed $\Lambda$ reduces to the identify matrix and $\mathbf{T}_n$ in equation (3.3) becomes the ABCD matrix $\mathbf{P}_n$ at frequencies $\omega_r$. Although the mathematical spaces for the LTI and LTP systems are different, the circuit based formalism suggests that space–time structures are formally equivalent to the well-known periodic structures. The infinite dimension of the LTP space and its translation operator highlights the *richness* in the spectrum (i.e. the eigen-functions) of space–time periodic structures, which eventually may lead to the possibility of breaking the physical limitations of LTI circuits. Ultimately, the properties of the LTI translation operator are dictated from the constraints of its basic circuit elements (stability, passivity, reciprocity and physical realizability), the properties of LTP operators hinge upon the fundamental characteristics of LTP elements, an area yet to be explored.

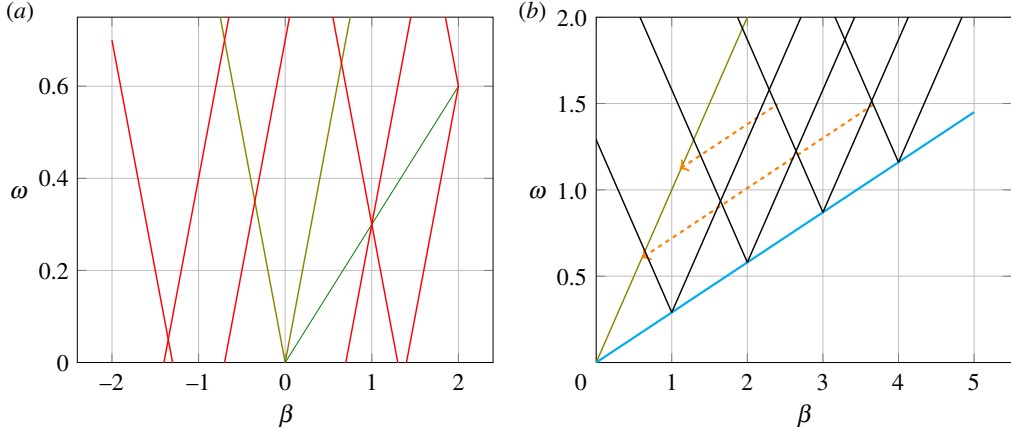

**Figure 2.** (*a*) A typical dispersion relation for a right-handed medium, when the modulation is very small (i.e. the bandgaps approach zero width). The main branches are distinguished by the olive lines (online). At a given frequency, shown by the blue dashed line, different modes are possibly excited. The modes are identified by the branch number $r$ with a subscript $\pm$ depending on the sign of $\beta$. (*b*) The dispersion relation of an RH TL when the modulation strength is infinitesimally small. Different modes at $\omega = 1.5$ are highlighted showing their corresponding counterparts on the main branch, albeit at different frequencies.

For any given frequency $\omega$ and in the limit of infinitesimal modulation strength, the **A**, **B**, **C** and **D** matrices become diagonal and the eigenmodes become the solution of the LTI system at frequencies $\omega_r$. The propagation constants $\beta_r$ satisfy the well-known dispersion relation [31,32]

$$\beta_r p = \cos^{-1}\left(\frac{A_r^r(\omega) + D_r^r(\omega)}{2}\right).$$

Furthermore, the eigenvectors become the Bloch waves. Figure 2*a* depicts the dispersion relation of an RH TL for an infinitesimal modulation strength. The modes are labelled as shown for different values of $r$. Note that for each $r$, there are two branches representing the solutions $\pm\beta_r$.

For the subsequent discussion, it is useful to introduce the harmonic shift operator $\mathcal{S}_\mathcal{U}$.

**Definition 3.3.** $\mathcal{S}_\mathcal{U}$ is a linear operator on $\Psi_n = [\cdots, \psi_{k-1}, \psi_k, \psi_{k+1}, \ldots]_n^t$ that has the following effect

$$(\mathcal{S}_\mathcal{U} \Psi_n)_k = (\Psi_n)_{k+1}.$$

i.e. $\mathcal{S}_\mathcal{U}$ shifts the vector $\Psi$ up by one position. Similarly $\mathcal{S}_\mathcal{D} \triangleq \mathcal{S}_\mathcal{U}^{-1}$ shifts the vector down by one position.

If $(\omega, \beta)$ is a solution of (3.3), then the following is a general property of arbitrary space time modulated structures (Proof in electronic supplementary material).

**Property 3.4.** Let $(\beta, \omega)$ be a solution to the eigenvalue problem (3.3), then $(\beta + l\hat{\beta}, \omega + l\hat{\omega})$, where $l \in \mathbb{Z}$ is also a solution. Moreover if $\Psi_n$ is the eigenvector at $(\beta, \omega)$ then $\mathcal{S}_\mathcal{U}^l \Psi_n$ is an eigenvector at $(\beta + l\hat{\beta}, \omega + l\hat{\omega})$.

**Corollary 3.5.** *All points $(\beta + l\hat{\beta}, \omega + l\hat{\omega})$ along the line $\omega' = \omega + \hat{v}(\beta' - \beta)$ are equivalent in the sense that the eigenvectors are all related by the harmonic shift operator $\mathcal{S}_\mathcal{U}$.*

The previous property should not be surprising. The change $\omega \to \omega + l\hat{\omega}$ and $\beta \to \beta + l\hat{\beta}$ is equivalent to re-numbering the harmonics. Such property can be exploited to understand how the different modes at a given frequency behave by restricting the analysis to the zero-order branches only. Consider for example the dispersion relation shown in figure 2*b*. Suppose one is interested in the modal behaviour at $\omega = 1.5$ a.u. Three modes are highlighted in the figure. Property 3.4 shows that the (2) eigenvector is a shifted copy of the (2') one. The (2') mode is at the intersection point of the $0_+$ and $1_-$ branches. As is already established when the modulation strength is relatively large, this point corresponds to the centre of a bandgap, where there is a strong interaction with the $-1$ harmonic [21,22]. Therefore, the voltage of this mode will contain non-zero entries only in the $-1$th and 0th

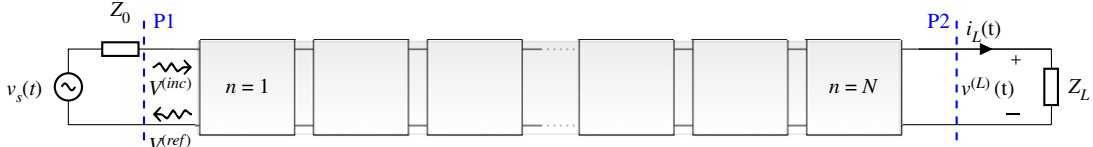

**Figure 3.** N unit cells of a space–time periodic structure connected to an input voltage source with a reference impedance $Z_0$ and a load of impedance $Z_L$.

locations. Property 3.4 implies that at $\omega = 1.5$ a.u., the (2) mode has a zero entry in the 0th location and hence does not couple to the excitation at $\omega$.

For the subsequent analysis, the $k$th eigenvalue and the corresponding eigenvector will be identified by the $(k)$ superscript. In general, the mode voltage $\mathcal{V}^{(k)}$ and current $\mathcal{I}^{(k)}$ are related via the *Bloch Admittance* matrix

$$\bar{\bar{\mathbf{Y}}} = [e^{i\beta^{(k)}p}\mathbf{e} - \mathbf{D}\boldsymbol{\mathit{\Pi}}]^{-1}\,\mathbf{C}\boldsymbol{\mathit{\Gamma}}.$$

The general solution is the superposition of all modes

$$v[n] = \sum_{k=1}^{\infty}\sum_{r=-\infty}^{\infty} a_k \mathcal{V}_r^{(k)}\, e^{i[\omega_r t - \beta_r^{(k)}np]} + c.c. \tag{3.6}$$

and

$$i[n] = \sum_{k=1}^{\infty}\sum_{r=-\infty}^{\infty} a_k \mathcal{I}_r^{(k)}\, e^{i[\omega_r t - \beta_r^{(k)}np]} + c.c. \tag{3.7}$$

# 4. Boundary value problem

Consider the structure in figure 3 that represents a generic space–time modulated structure that is connected to a source and load. The incident ($v^{(\mathrm{inc})}$, $i^{(\mathrm{inc})}$) and reflected ($v^{(\mathrm{ref})}$, $i^{(\mathrm{ref})}$) waves appear on a transmission line of characteristic impedance $Z_0$ (usually a 50 $\Omega$ microstrip or coaxial TL) that connects the structure to a voltage source $v_s(t)$. At the input side, $v_s(t) = 2v^{(\mathrm{inc})}(t)$. At the input of the first unit cell ($x = 0$), the boundary conditions are imposed by requiring the voltage and current to be continuous at the input and output ports. Therefore, for a sinusoidal excitation $v_s(t) = V_s\cos(\omega t + \phi)$ and a load impedance $Z_L$, one gets

$$\sum_{k=1}^{\infty} \mathcal{V}_r^{\mathrm{inc1},k} a_k = V^{\mathrm{inc}}\delta_r^0, \quad r = \cdots, -2, -1, 0, 1, 2, \ldots, \tag{4.1}$$

where $\mathcal{V}_r^{\mathrm{inc1},k} \triangleq (\mathcal{V}_r^{(k)} + Z_0\mathcal{I}_r^{(k)})$ is the contribution of the $k$th mode to the wave incident on P1 and $V^{\mathrm{inc}} = V_s\,e^{i\phi}/2$. Equation (4.1) shows that the coefficients $a_k$ are such that the net effect of the branches is balanced with the excitation at frequency $\omega$ and they *destructively* interfere at any other harmonics. At the load side,

$$\sum_{k=1}^{\infty} a_k \mathcal{V}_r^{\mathrm{inc2},k}\, e^{-i\beta_r^{(k)}Np} = 0, \tag{4.2}$$

where $\mathcal{V}_r^{\mathrm{inc2},k} \triangleq (\mathcal{V}_r^{(k)} - Z_L\mathcal{I}_r^{(k)})/2$ is the wave reflected from the load $Z_L$ when the output port is terminated in $Z_L$. For the remainder of the manuscript, it is assumed that $Z_L = Z_0$ (i.e. the structure is terminated in the reference impedance $Z_0$). Therefore, (4.2) implies that the $a_k$ coefficients are the ones that result in a null reflection from the load at all harmonics.

In practical applications and away from the luminal regime, only a limited number $N_H$ of harmonics are significant. For convenience, we consider $N_H$ to be an odd number $2\mathcal{N}_s + 1$, where $\mathcal{N}_s = 0, 1, 2, \ldots$. This selection allows the symmetric inclusion of harmonics from $-\mathcal{N}_s$ to $\mathcal{N}_s$. Furthermore, we consider the number of branches to be $2N_H$ to account for forward and backward waves. For each harmonic, the truncated versions of (4.1) and (4.2) provide two equations in the $2N_H$

coefficients $a_k$. Taking all $N_H$ harmonics into account, we end up with a system of $2N_H$ equations in $2N_H$ unknowns that can be written as

$$*** \begin{pmatrix} \mathcal{V}^{\text{inc1},1}_{-\mathcal{N}_s} & \mathcal{V}^{\text{inc1},2}_{-\mathcal{N}_s} & \cdots & \mathcal{V}^{\text{inc1},2N_H}_{-\mathcal{N}_s} \\ \vdots & \vdots & \vdots & \vdots \\ \mathcal{V}^{\text{inc1},1}_{0} & \mathcal{V}^{\text{inc1},2}_{0} & \cdots & \mathcal{V}^{\text{inc1},2N_H}_{0} \\ \vdots & \vdots & \vdots & \vdots \\ \mathcal{V}^{\text{inc1},1}_{\mathcal{N}_s} & \mathcal{V}^{\text{inc1},2}_{\mathcal{N}_s} & \cdots & \mathcal{V}^{\text{inc1},2N_H}_{\mathcal{N}_s} \\ \mathcal{V}^{\text{inc2},1}_{-\mathcal{N}_s} & \mathcal{V}^{\text{inc2},2}_{-\mathcal{N}_s} & \cdots & \mathcal{V}^{\text{inc2},2N_H}_{-\mathcal{N}_s} \\ \vdots & \vdots & \vdots & \vdots \\ \mathcal{V}^{\text{inc2},1}_{0} & \mathcal{V}^{\text{inc2},2}_{0} & \cdots & \mathcal{V}^{\text{inc2},2N_H}_{0} \\ \vdots & \vdots & \vdots & \vdots \\ \mathcal{V}^{\text{inc2},1}_{\mathcal{N}_s} & \mathcal{V}^{\text{inc2},2}_{\mathcal{N}_s} & \cdots & \mathcal{V}^{\text{inc2},2N_H}_{\mathcal{N}_s} \end{pmatrix} \begin{pmatrix} a_1 \\ \vdots \\ a_{\mathcal{N}_s+1} \\ \vdots \\ a_{N_H} \\ a_{N_H+1} \\ \vdots \\ \vdots \\ \vdots \\ a_{2N_H} \end{pmatrix} = \begin{pmatrix} 0 \\ \vdots \\ V^{(\text{inc})} \\ \vdots \\ 0 \\ \vdots \\ \vdots \\ \vdots \\ \vdots \\ 0 \end{pmatrix} \tag{4.3}$$

From the solution of (4.3), the transmission coefficient of the $r$th harmonic $S_{21}^{(r,0)}$ can be calculated.

# 5. Results and discussion

To verify and demonstrate the use of the machinery developed in §§3 and 4, we will apply the framework to analyse two main structures. The first is a space–time periodic CRLH TL. Such an idealistic model allows a thorough analysis of the propagation behaviour that can be compared with state space time domain simulations. Next, we use the framework to reproduce and give insight into the non-reciprocial behaviour observed on a nonlinear right-handed transmission line (NL RH TL) that has been manufactured in our lab. Although the analysis carried out below is not meant to be exhaustive, it provides a useful and a systematic procedure to describe complex space–time periodic structures.

## 5.1. Composite right-left handed space–time modulated TL

The CRLH consists of $N = 40$ unit cells such as the one shown in figure 4, where the right-handed capacitance $C_R$ is space–time modulated. The first unit cell is connected to a source of impedance 50 $\Omega$. The load is also assumed to be 50 $\Omega$. KCL and KVL along with the current and voltage relations in the time domain are used to derive a state space model (SSM) of the circuit that can be written as

$$\dot{\mathbf{x}} = \mathbf{A}(t)\mathbf{x} + \mathbf{B}(t)u,$$

where $\mathbf{x}$ is an $N \times 1$ vector that stores the state variables (current in inductors and voltages across capacitors), $\mathbf{A}$ is a $N \times N$ matrix, $\mathbf{B}$ is a $N \times 1$ vector that connects the input excitation to the states. The unit cell shown in figure 4 can be divided into three sub-units: (1) the LTI series impedance $Z_{\text{se}} \triangleq i\omega L_R - i/\omega C_L$, (2) shunt admittance $1/\omega L_L$, and (3) the LTP admittance $\tilde{\mathbf{Y}}_R$. Hence, the ABCD matrix of the unit cell can be formed by cascading its three sub-units. Therefore, the different eigenvalues $e^{i\beta^{(k)}p}$ and the corresponding eigenvectors $(\mathcal{V}^{(k)}, \mathcal{I}^{(k)})^t$ are determined through the use of (3.3). Subsequently, when the TL is excited by a sinusoidal source of frequency $\omega$, the boundary value problem (4.3) is solved and the modes coefficients $a_k$ are computed.

The LTP dispersion relation when $C_R$ is sinusoidally modulated, i.e. $C_R = C_{R0}[1 + M\cos(\hat{\omega}t - \hat{\beta}np)]$ is obtained from the eigenvalues as depicted in figure 5a. The LTI dispersion relation (when $M = 0$) is superimposed to highlight the right-hand (RH) and left-hand (LH) regions. The LH region is in the low-frequency range, frequencies below 1 a.u., where the phase and group velocities are opposite [33,34]. For a balanced operation, the series and shunt resonances were both set to unity [34]. Figure 5b,c shows a close up view of the dispersion relation in the RH and LH regions, respectively. Different modes are highlighted and labelled. It is worth noting that whenever two branches appear to intersect, two complex conjugate propagations constants ($\gamma = -i\beta$) are generated. The point of intersection represents the centre of a bandgap, where strong coupling with time harmonics may be significant. For instance consider figure 5b, where points 7 and 8 represents two eigenvectors that have two complex conjugate propagation constants.

The eigenvalues, when the frequency is at the centre of the RH BG ($f = 1.5$ a.u.), are depicted in figure 6a. Unlike the first four, the higher eigenvalues result in complex conjugate propagation constants. This is not

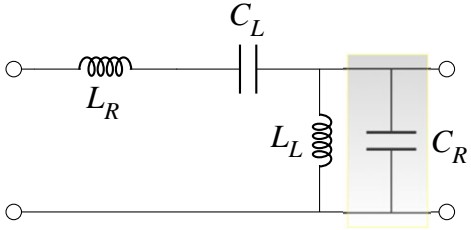

**Figure 4.** Unit cell of a space–time modulated CRLH TL. The right-handed capacitance $C_R$ is modulated as a travelling wave $C_R = C_{R0}[1 + M\cos(\hat{\omega}t - \hat{\beta}n)]$.

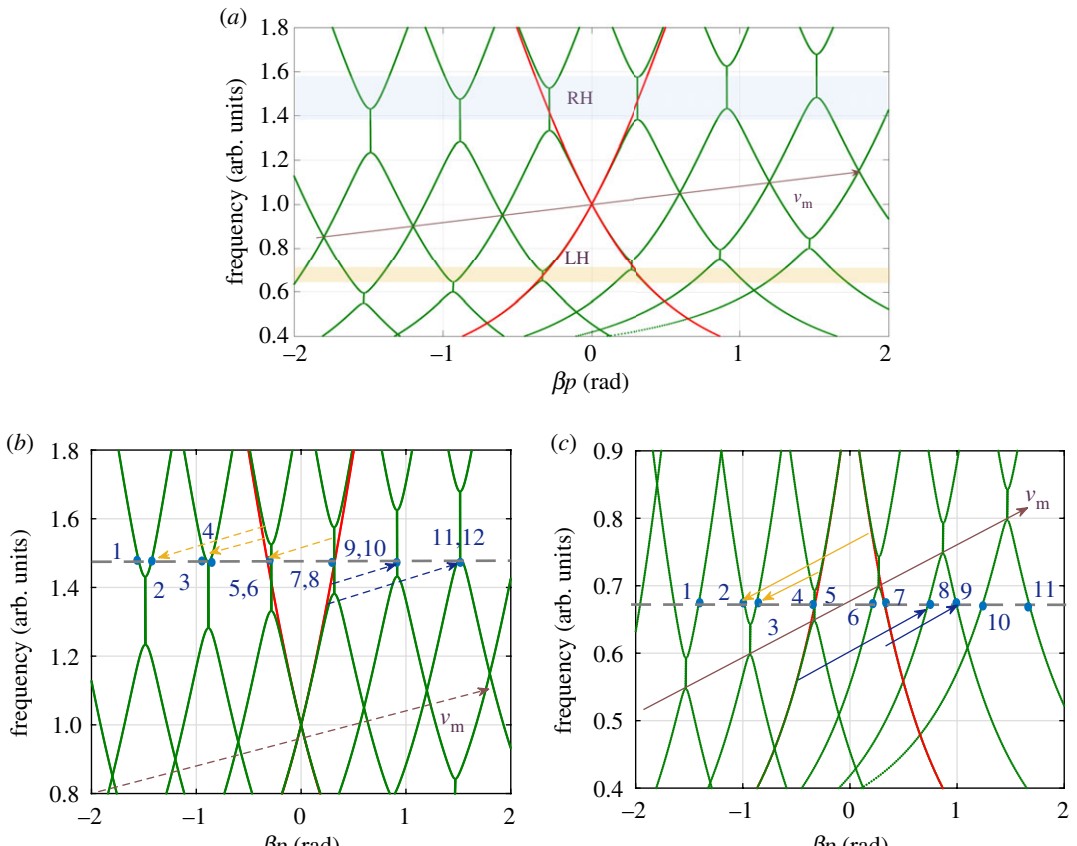

**Figure 5.** The dispersion relation of a CRLH TL modulated by a forward travelling wave. $M = 0.4$, $\omega_{se} = \omega_{sh} = 1$ a.u., $\omega_{RH} = 2.5$ a.u. (a) Dispersion relation of a CRLH TL modulated by a forward travelling wave. a.u. stands for arbitrary unit. (b) Zoomed view of the RH region. (c) Zoomed view of the LH region.

surprising since they correspond to points inside BGs as figure 5b shows. For any given eigenvalue, the magnitude of the components of the corresponding eigenvector is plotted in figure 6b. For a given eigenvector (mode), the y-axis represents the strength of the rth harmonic. According to (3.6), the waveform inside the space–time periodic structure is the linear superposition of the different eigenvectors. Figure 6d plots the magnitude of the expansion coefficient $a_k$. Clearly, the wave behaviour is dominated by the 8th eigenvector, which corresponds to one of the modes inside the BG of the main branch as illustrated in figure 5b. For such modes, figure 6b shows that the 0th and −1th harmonics are dominant and of the same order of magnitude, inferring a strong interaction between the fundamental and its −1th harmonic. Additionally, there is a small contribution coming from the 6th and 9th modes. The figure also shows that the 6th and 9th modes have significant components at the $0, +1$ and $0, -1$ enteries, respectively. Note that the behaviour of the two modes can also be deduced using property 3.4. Indeed, the equivalent mode of the 6th (9th) one on the main branch is inside the first BG. Hence the equivalent mode has non-zero values at the −1 and 0 entries. Since the eigenvector of the 6th (9th) mode is a down (up) shifted copy, it has non-zero values at the $+1, 0$ ($-1, 0$) entries, in agreement with figure 6b.

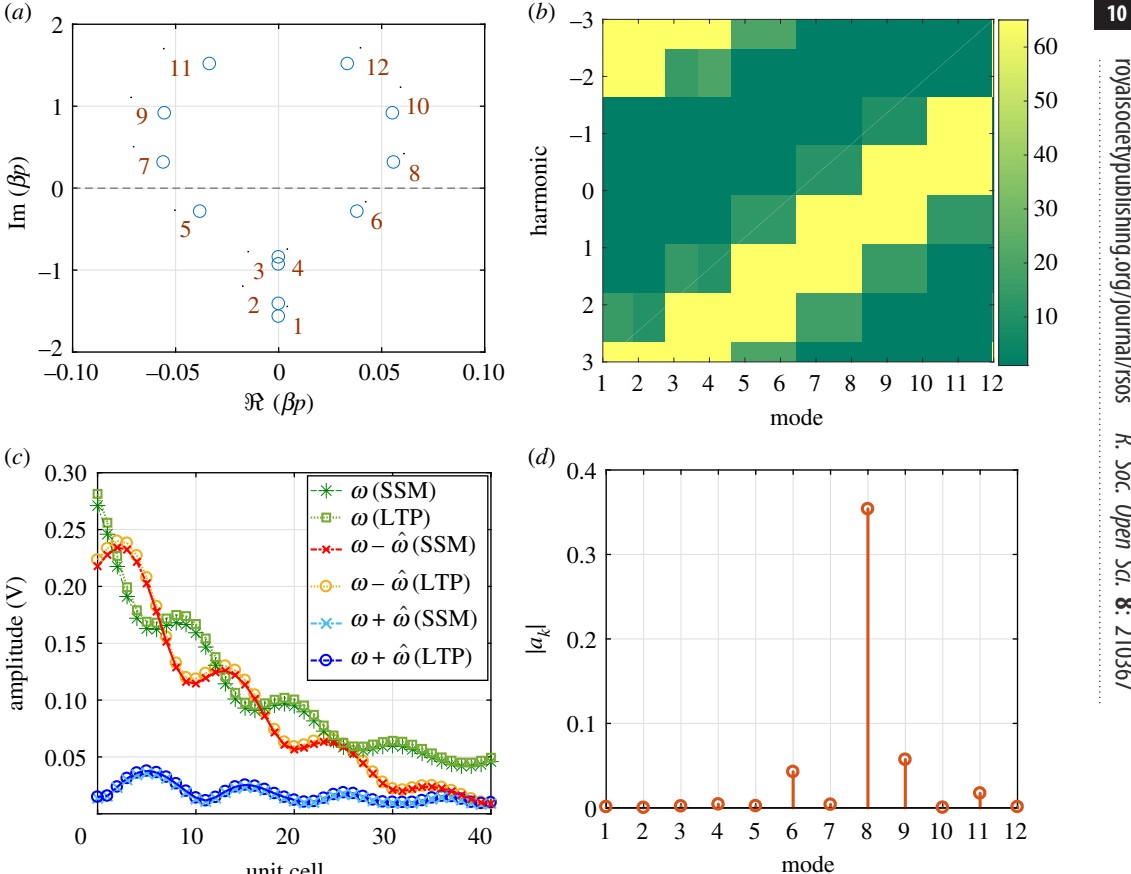

**Figure 6.** Computed eigenvalues, eigenvectors and waveforms at the centre of the RH BG of the LTP CRLH TL, when $M = 0.4$. (a) Real and imaginary of $\beta p$ for different modes. (b) Magnitude of different components of eigenvectors $\mathcal{V}^{(k)}$. (c) Amplitude of waves at signal frequency $\omega$, the $\omega - \hat{\omega}$ and $\omega + \omega$ harmonics. (d) Magnitude of $a_k$.

To assess how accurately the LTP approach can predict the wave behaviour inside the structure, the waveform at the middle of the RH BG, at the three frequencies $\omega$, $\omega - \hat{\omega}$ and $\omega + \hat{\omega}$ are calculated using (3.6) and compared with the solution of the SSM. The time domain data obtained from the SSM simulation is transformed to the frequency domain, where the frequencies of interest are isolated. Figure 6c reports the amplitude of the three harmonics. As shown, there is an excellent agreement between LTP and SSM. Additionally, the amplitude of the main component at $\omega$ rapidly decreases as the wave penetrates into the structure, where it is scattered (mainly) in the $-1$ harmonic back to the source. Furthermore, there is a non vanishing contribution, coming from the $+1$ harmonic, as a result of the excitation of the 6th mode.

The same procedure is repeated but for $\omega$ at the centre of the LH BG (figure 5c). Unlike the RH BG, the incident and modulating waves are contra-directional. This is due to the left handedness of the CRLH in this regime. Therefore, the incident wave scatters in the $\omega + \hat{\omega}$ (blue shifted) as figure 7c highlights. The scattering, however, is not as strong as in the RH BG case. This is due to the smaller magnitude of the real part of the eigenvalue (figure 7a) and witnessed by the slight reduction of the amplitude of the fundamental component (figure 7c). It is expected that a modulation of $L_L$ or $C_L$ will result in a wider LH BG. Note also that unlike the dominant mode in the RH BG (mode 8, figure 6a), the dominant mode in the LH BG has a negative phase velocity as witnessed by the imaginary part of $\beta p$ (mode 7, figure 7a).

Finally, the transmission coefficient is calculated over a wide frequency range that includes both the RH and LH BGs. Figure 8a,b presents the results, when the incident and modulation waves are co- and contra-directional, respectively. The figures show that LTP-based calculations are in a very good agreement with SSM. Furthermore, space–time modulation has the effect of attenuating the transmitted signal ($-15$ dB for the LH BG and $-30$ dB for the RH BG), compared to almost 0 dB when modulation is absent.

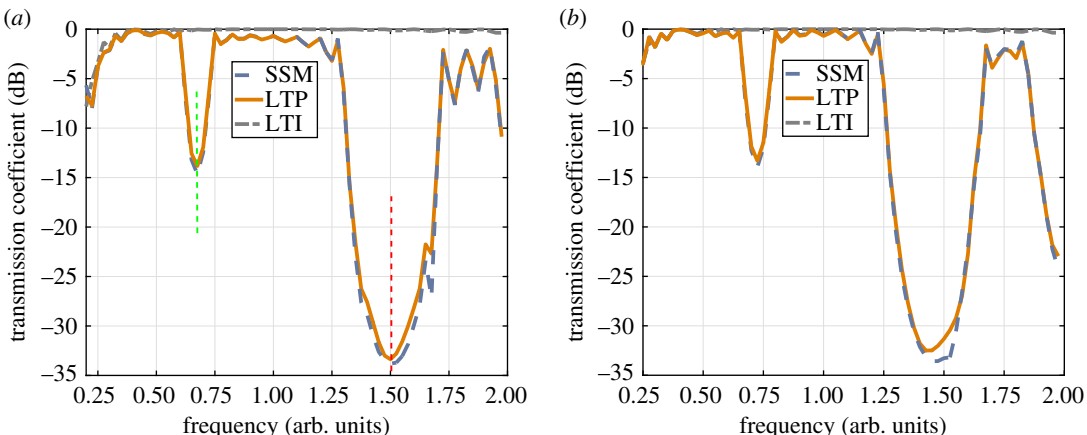

**Figure 7.** Computed eigenvalues, eigenvectors and waveforms at the centre of the LH BG of the LTP CRLH TL, when $M = 0.4$. (a) Real and imaginary of $\beta p$ for different modes. (b) Magnitude of different components of eigenvectors $\mathcal{V}^{(k)}$. (c) Amplitude of waves at signal frequency $\omega$, the $\omega - \hat{\omega}$ and $\omega + \hat{\omega}$ harmonics. (d) Magnitude of $a_k$.

**Figure 8.** Transmission coefficient calculated for a space–time modulated CRLH, with a modulation depth $M = 0.8$ using the LTP formalism and brute force time domain computation. (a) Forward modulation (i.e. $\hat{\beta} > 0$). (b) Backward modulation (i.e. $\hat{\beta} < 0$).

## 5.2. Analysis of a nonlinear right-handed transmission line (NLRHTL)

A modulating sinusoid $v_m(t)$ and an input signal $v_s(t)$ were applied to a nonlinear right-handed transmission line (NLRHTL) built from 20 unit cells, as shown in figure 9a. The inputs $v_m$ and $v_s$ are combined using a directional coupler as highlighted. Each unit cell consists of a $p \approx 6.5\,\mathrm{mm}$ long microstrip loaded at its centre by a varactor (M/A-COM, MA46H120). The circuit is etched on a 25

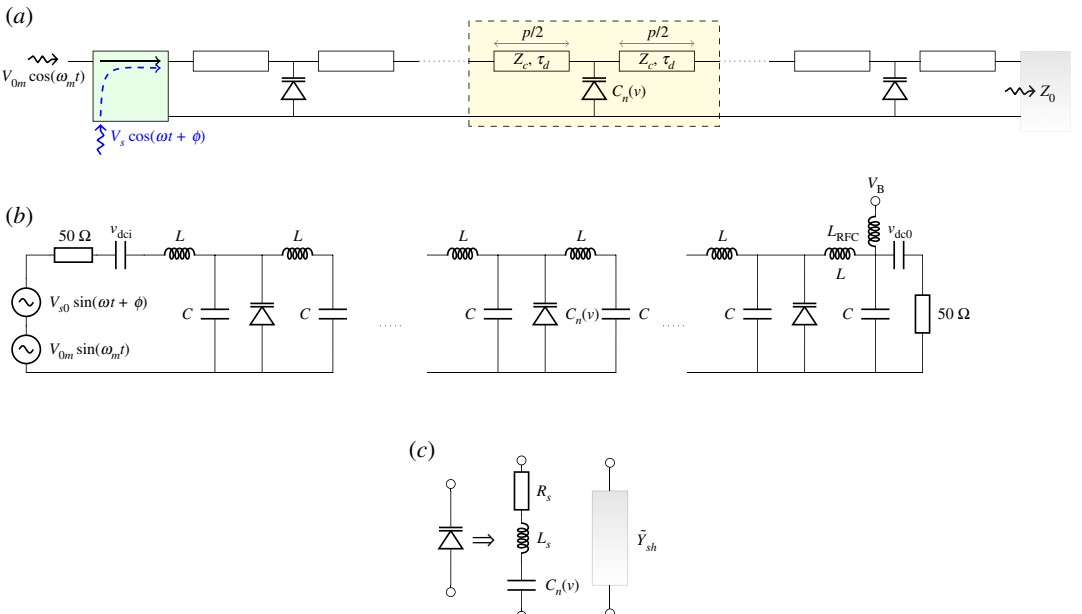

**Figure 9.** (a) Schematics of a synthesized NLRHTL built from microstrip sections that are loaded by shunt varactors. The length of the unit cell $p$ is approximately 6.5 mm, the microstrip dimensions and substrate are such that $Z_c = 64.26\ \Omega$, $\tau_d \approx 27.2$ ps. (b) The microstrips are modelled by lumped LC sections. Additionally, DC blocking capacitors are included. Furthermore the bias voltage $V_B$ is applied via a ferrite bead that is modelled by a high inductance $L_{RFC}$. (c) The varactor is modelled by a series RLC circuit. $R_s$ represents the ohmic losses, $L_s$ the parasitics due to bond-wires and soldering, and $C(u_{pn})$ is the nonlinear capacitance value.

mil thick Rogers RO3010 substrate. The varactors are bonded in place using H20E conductive epoxy. For more details about the transmission line and the experimental set-up, refer to the electronic supplementary material.

The capacitance of the varactor $C_v$ depends on the voltage across its terminals $u(t) = u_m(t) + u_s(t)$, where $u_m(t)$ and $u_s(t)$ are the voltages due to the modulating input and signal, respectively. The current through the varactor is given by

$$i(t) = C_n(u)\frac{\mathrm{d}u}{\mathrm{d}t}.$$

Since $C_v(u) = C_v(u_m + u_s) \approx C_n(u_m) + u_s \mathrm{d}C_n/\mathrm{d}t|_{u_m}$, it can be shown that the current $i_s(t)$ due to the signal excitation is

$$i_s(t) = \frac{\mathrm{d}}{\mathrm{d}t}C_n(u_m)u_s(t),$$

where $C_n(u_m)$ is the capacitance evaluated at $u_m(t)$, which is periodic with frequency $\hat{\omega}$. Hence, the system is linearized about the limit cycle steady state [35]. Figure 9c shows the varactor's equivalent circuit. $R_s$ models the ohmic losses in the semiconductor bulk, contact and bondwires, $L_s$ accounts for the inductance of the bondwires, and $C$ represents the varactor capacitance. The different circuit parameters were extracted from measuring the S parameters at different bias voltage and fitting the response via the use of the vector fitting technique [36].

At low frequency, the microstrip line can be described by lumped circuits as in figure 9b. The $p/2$ microstrip line section is modelled as a lumped LC network, such that $L = \tau_d Z_c$ and $C = \tau_d/Zc$, where $\tau_d$ and $Z_c$ are the delay and characteristic impedance of the microstrip, respectively. The lumped circuit approximation allows the convenient representation of the NL RH TL in an SSM form. In this case, the biasing circuit and blocking capacitances can be included as in figure 9b.

### 5.2.1. Dispersion relation

As a first step, the LTI dispersion relation of the structure is extracted from measuring the small signal S parameters for different bias voltages and compared to the circuit models. Figure 10 shows the dispersion relation curves of four bias voltages. Clearly, both the lumped and distributed circuit models are in agreement with measurement, confirming the validity of the lumped circuit model.

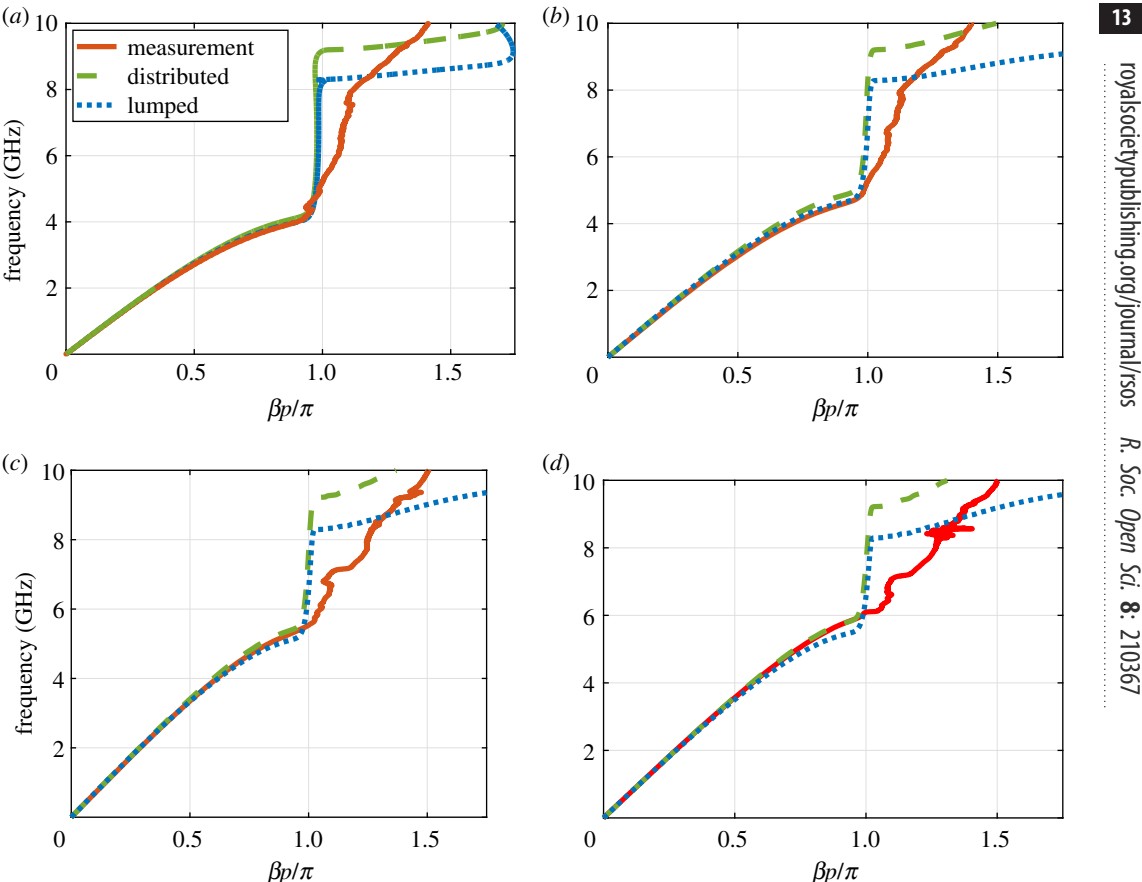

**Figure 10.** LTI dispersion relation of the NLRHTL for different bias voltages. ($a$) $V_B = 0$ V, ($b$) $V_B = 1$ V, ($c$) $V_B = 2$ V, ($d$) $V_B = 3$ V.

In the presence of the modulating signal with frequency $\hat{\omega}$, the varactor capacitance $C_n$ becomes periodic with a period of $2\pi/\hat{\omega}$. Therefore, it can be expanded in Fourier series

$$C_n(t) = \sum_{r=-\infty}^{+\infty} \tilde{C}_r\, e^{ir\hat{\omega}t}.$$

Since the amplitude of modulation is large ($u_m \gg u_s$), the DC capacitance $\tilde{C}_0$ may deviate from the small signal value. In the subsequent analysis, the DC and first harmonic only ($\tilde{C}_0$ and $\tilde{C}_1$) will be considered. They are calculated from a time domain simulation of NLRHTL. The system differential equations are solved to compute the voltages $u_m$ across the different varactors. Consequently, $\tilde{C}_0$ and $\tilde{C}_1$ are calculated from the Fourier transform of $C(u_m)$. Figure 11$a$ shows the computed spectrum of $C_v$ across the 10th varactor. The DC capacitance $\tilde{C}_0$ has increased from approximately 1.2 pF to 1.35 pF. The modulation strength $M \triangleq \tilde{C}_1/\tilde{C}_0 \approx 0.2$ when the excitation is approximately $10 - 15$ dBm. Figure 11$b$ demonstrates how $\tilde{C}_0$ and $\tilde{C}_1$ change from one unit cell to the other. Although not constant, we will assume that both $\tilde{C}_0$ and $\tilde{C}_1$ are constants and fixed to their average values. This assumption allows the application of the LTP formalism and captures the main essence of the system, as will be shown below.

The unit cell can be represented by the block diagram in figure 12. Hence the ABCD matrix of the unit cell is $\mathbf{T} = \mathbf{T}_{\text{LTI}}\mathbf{T}_{\text{LTP}}\mathbf{T}_{\text{LTI}}$. For the microstrip, the ABCD parameters are identical to the LTI counterpart, but calculated at each harmonic frequency $\omega_r$.

The LTP block $\mathbf{T}_{\text{LTP}}$ represents the ABCD parameters of the shunt varactor, which is modelled by a shunt time periodic admittance $\tilde{\mathbf{Y}}_{\text{sh}} = (\mathbf{Z}_{\text{se}} + \tilde{\mathbf{Y}}^{-1})^{-1} = \tilde{\mathbf{Y}}(\mathbf{e} + \mathbf{Z}_{\text{se}}\tilde{\mathbf{Y}})^{-1}$. The second term in the last expression is the inverse of a tridiagonal matrix and can be computed using closed-form expressions as in [37].

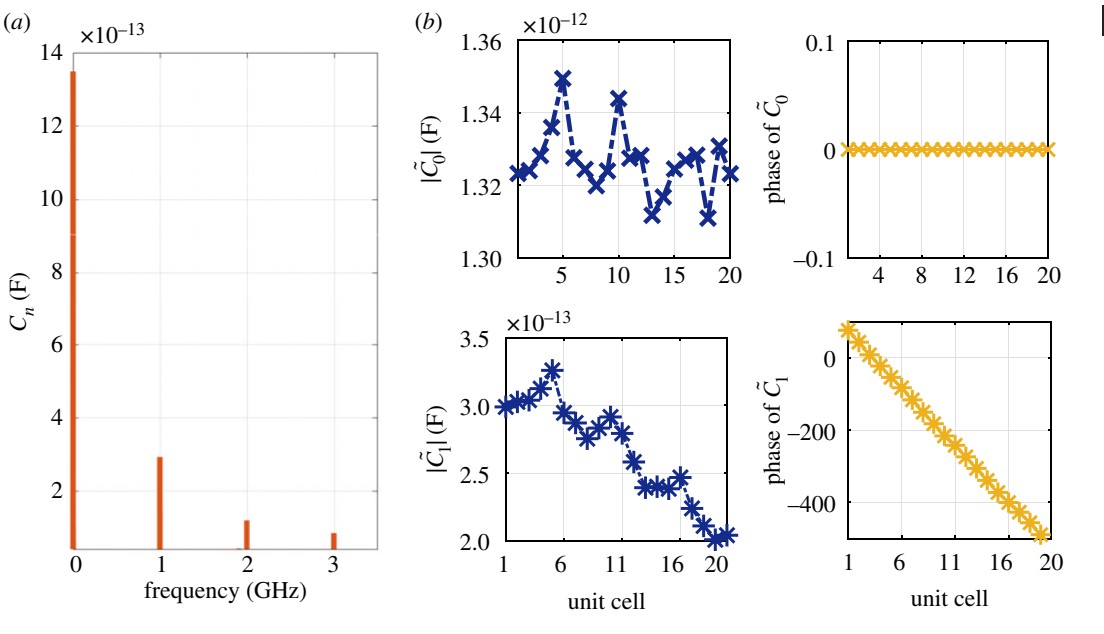

**Figure 11.** Varactor capacitance for the NL RH TL. (*a*) Spectrum of $C_n$ at the 10th unit cell, when $f_m = 1\,\text{GHz}$ and $V_m = 15\,\text{dBm}$. (*b*) Capacitance of varactor at each unit cell, obtained from the state space model.

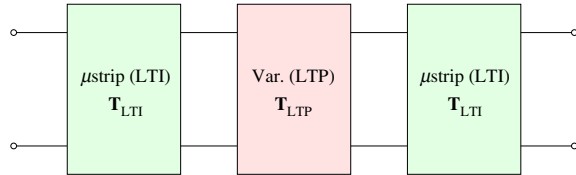

**Figure 12.** The unit cell of the modulated TL modelled as the cascade of three sections: two LTI sections representing the microstrip lines and an LTP section describing the terminals behaviour of the varactor.

The speed of modulation $\hat{v}$ is determined from the phase $\phi$ of $\tilde{C}_1$, where

$$\hat{v} = 2\pi f_m p \left| \frac{\Delta n}{\Delta \phi} \right|,$$

which is expected to *slightly* deviate from the TL LTI speed. For a given modulation frequency $f_m$ and strength $M$, the dispersion relation can be determined from the solution of (3.3). Figure 13 depicts the dispersion relation for $f_m = 1\,\text{GHz}$, and when the modulation propagates in forward (figure 13*a*) and backward (figure 13*b*) directions. As shown, $\hat{v}$ is very close to the LTI speed, suggesting that the LTP system is in the sonic regime [21,38].

To explore the interaction between the different modes and how they contribute to the overall propagation, consider the situation where the modulation and signal are co-directional. Using 14 modes, the eigenvectors are calculated as reported in figure 14*b*. The TL is excited with a sinusoidal signal of frequency $f = 2.82\,\text{GHz}$. As will be shown later, at this frequency, maximum non-reciprocity is observed. The plot shows the magnitude of the components of each eigenvector normalized to its maximum value. Modes of interest are the ones that strongly couple with the input excitation; hence they have significant components at $\omega$ (or the 0th harmonic as highlighted in figure 14*b*) and can potentially be excited. Additionally, the BVP (4.3) is invoked to compute the different $a_k$ values that in turn determine the strength of the excited modes as figure 14*d* shows. The waveforms at different frequencies are the superposition of the corresponding eigenvectors as presented in figure 14*c* and confirmed with SSM in figure 14*a*. Furthermore, figure 14*e* demonstrates that the waveforms can be approximated by the dominant eigenmodes (i.e. the ones that couple with the input excitation such that their expansion coefficients $a_k$ are non-vanishing). The signal at $\omega$ is significantly reduced at the output due to the interaction with its harmonics, mainly the −1 harmonic. The absence of bandgaps in the dispersion plots in figure 13 suggests that this type of interaction is passive in nature (i.e. $\beta$ is

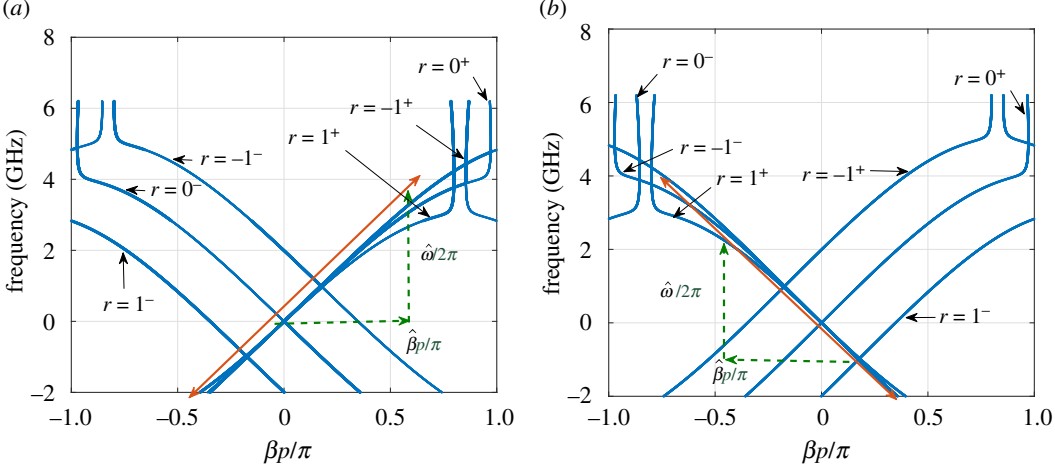

**Figure 13.** LTP dispersion relation of the NLRHTL, when $f_m = 1$ GHz. (a) Forward modulation. (b) Backward modulation.

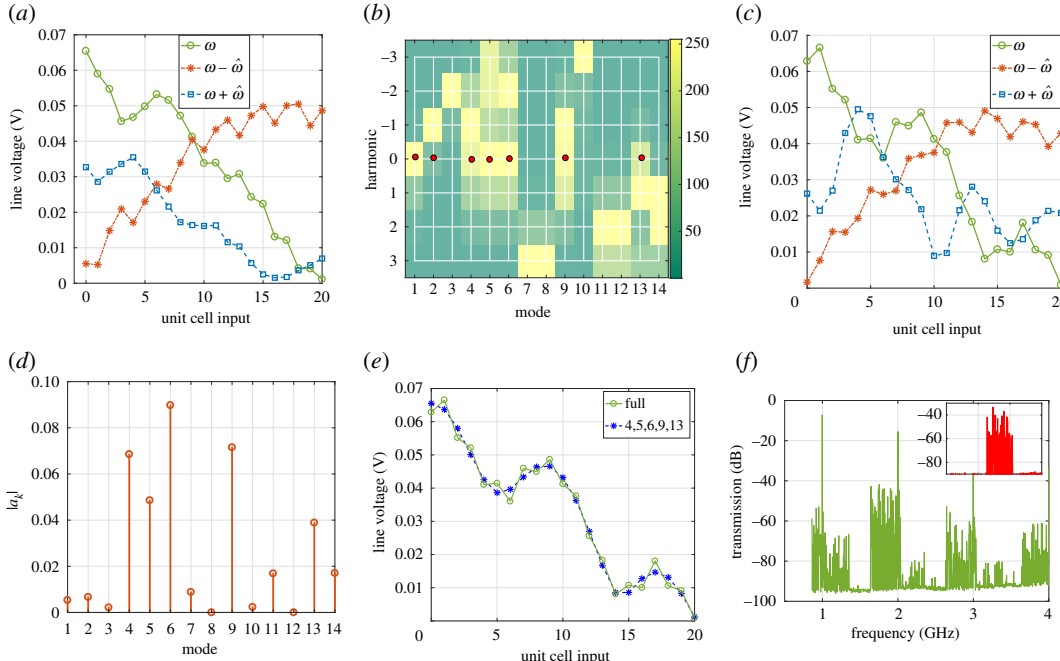

**Figure 14.** Amplitude of frequency components computed at $\omega$, $\omega - \hat{\omega}$ and $\omega + \hat{\omega}$ for the $N = 20$ NL RH TL, using (a) SSM, (c) LTP. (b) Magnitude of the first 14 modes. The modes strongly couple to the excitation are highlighted with the (red: online) dots. (d) The magnitude of the expansion coefficient $a_k$ of the different modes. (e) The voltage at $\omega$ calculated using the full 14 modes and compared with the one calculated using the relevant modes only. (f) Measured spectrum of the NL RH TL, where $\omega$ was allowed to sweep slowly over a frequency range around the dip in $S_{21}$. The inset shows the spectrum when the modulation is removed (i.e. pump excitation turned off).

imaginary) [39]. Such an implication can be demonstrated by plotting $\beta$ in the complex plane as in figure 15a. Note that the excited modes, as witnessed by the values of $|a_k|$ in figure 14d, have imaginary propagation constants. Additionally, an SSM computation of the same TL, but with an $N = 100$ unit cell is performed and the results are reported in figure 15b. Up to the 20th cell, the wave behaviour and interaction between harmonics resemble that of an $N = 20$ unit cells shown in figure 14a,c,e, where energy is mainly transferred from the fundamental to its $-1$ harmonic. Nevertheless, for the subsequent stages, up to the 60th cell, energy is pumped back to the fundamental and the amplitude of the fundamental harmonic increases; a typical behaviour of a passive interaction.

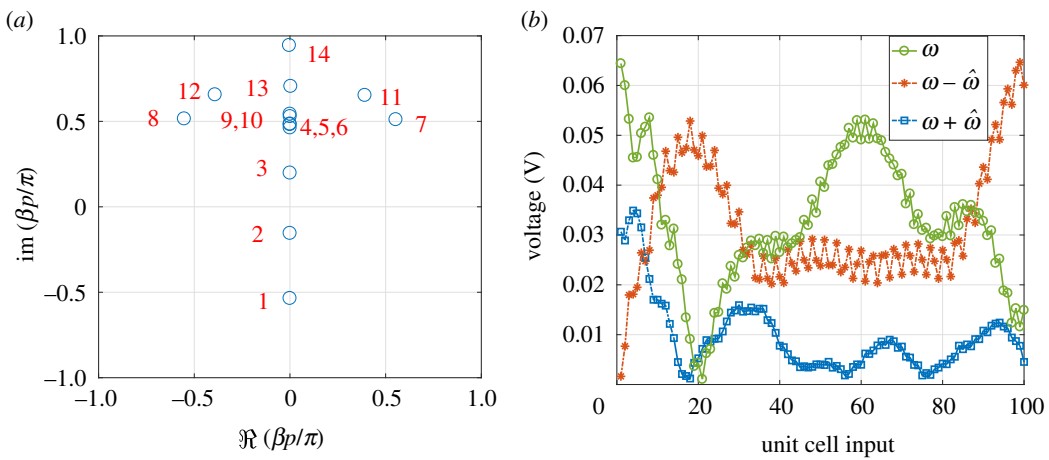

**Figure 15.** (a) Eigenvalues of the LTP circuit of NL RH TL. (b) Amplitudes of frequency components $\omega$, $\omega - \hat{\omega}$ and $\omega + \hat{\omega}$ computed using SSM at the dip in $S_{21}$, when $f_m = 1$ GHz and for $N = 100$ stages.

**Figure 16.** Modulation and signal are contra-directional. (a) Eigenvalues. (b) The magnitude of the expansion coefficients $a_k$. (c) Dispersion relation. (d) Magnitude of the components of the eigenvectors.

The strong interaction between the fundamental and its −1 harmonic is apparent from the measured output spectrum (figure 14f). Here, the input port was fed by an RF source that was swept over a frequency range around 2.82 GHz and the output of the spectrum was measured by a spectrum analyzer. The spectrum shows that once the modulation is turned on, the interaction is mainly with

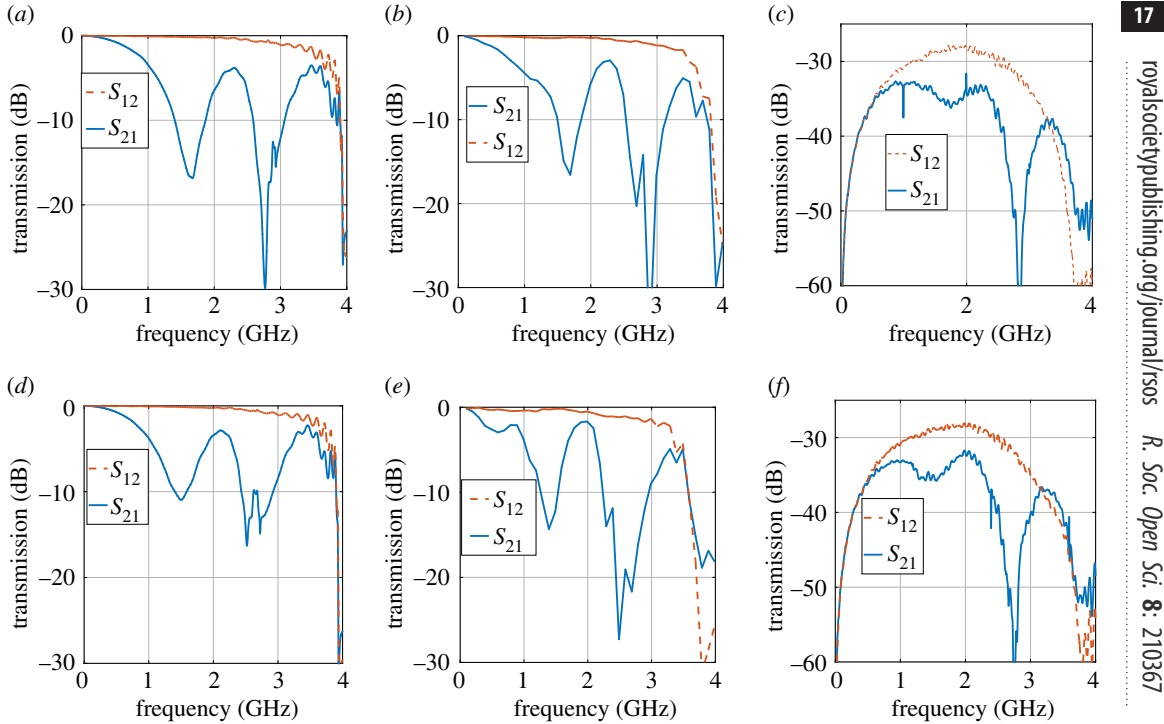

**Figure 17.** Transmission in the forward and backward directions. (*a*) LTP: $f_m = 1$ GHz. (*b*) SSM: $f_m = 1$ GHz. (*c*) Measurement: $f_m = 1$ GHz. (*d*) LTP: $f_m = 1.2$ GHz. (*e*) SSM: $f_m = 1.2$ GHz. (*f*) Measurement: $f_m = 1.2$ GHz.

the −1 harmonic. Note that modulation and its higher harmonics (1, 2 and 3 GHz) appear as spikes in the measured spectrum.

When the modulation and signal are contra-directional, as figure 16*a* demonstrates, the eigenvalues are generally different from those calculated above. The dispersion relation shows an increase in the separation between the forward branches as in figure 16*c*. Hence, the incident wave is expected to strongly couple to the main branch, labelled by the mode number 14. Note that other higher modes, for instance mode 15, are wrapped back to the negative side once $\beta p$ exceeds $\pi$. It is worth noting from the computed eigenvectors (figure 16*d*) that the 9th and 10th modes have a significant component at the 0th harmonic. However, due to the increased separation between the branches in the forward direction such modes are not excited. Therefore, one may conclude that when the signal and modulation are contra-directional the propagation is basically that of the LTI system. Indeed, the calculated $a_k$ coefficients (figure 16*b*) show that coupling is mainly with the 14th mode. Therefore, the mode couples with the forward main branch and the structure appears to be transparent in this mode of operation.

### 5.2.2. Transmission coefficient

Finally, the modes are superimposed and the transmission coefficient is calculated for both the co- and contra-directional modes of operations. The transmission coefficient at the fundamental frequency is calculated using SSM and the process is repeated over the 0–4 GHz range. Figure 17 shows that as a consequence of space–time modulation and the asymmetric interaction between harmonics in the forward and backward directions, strong non-reciprocity between the forward and backward propagation arises. As has been shown, this is due to the *passive* interaction between the fundamental mode and its lower harmonic when the modulation and signal are co-directional. For an input frequency of 2.82 GHz, the coherent length is 20 unit cells long. Hence, maximum energy is transferred to the −1th harmonic, reducing the signal at the output port. In the opposite direction, however, the distances between the forward branches are widened and the effect of modulation is negligible. Figure 17*c,f* reveals that such non-reciprocal behaviour demonstrates itself in the measured scattering parameters. The baseline, however, is reduced by approximately 30 dB due to the presence of the directional coupler.

# 6. Conclusion

The time periodic circuit theory was exploited to derive some of the properties of the infinite-dimensional spatial translation operator of space–time modulated circuits. The modal behaviour of a generic space–time periodic structure can be explained from the solution of the system eigenvalue problem. Additionally, we showed that the translation operator guarantees that solutions are invariant under spatial translation. Furthermore, it was shown that all points in the $(\beta, \omega)$ plane parallel to the modulation velocity $\hat{v}$ are equivalent in the sense that the eigenvectors are related by a shift operator. The waveforms inside the space–time periodic circuit and the time periodic scattering parameters were determined through the expansion of the total solution in terms of the eigenmodes, and after imposing the suitable boundary conditions. Two examples were discussed. In the first, a space–time modulated CRLH TL was studied using the developed approach and compared with time domain simulation. In the second example, the non-reciprocal behaviour observed on a nonlinear TL was explained. This was made possible via the extraction of circuit parameters from measurements that were then used to predict the wave behaviour inside the TL and its effect on the terminal properties. It was shown that the passive interaction between different harmonics resulted in an observed non-reciprocal behaviour, where the difference between forward and backward transmission coefficients $S_{21} - S_{12}$ can be significant. The frequencies at which non-reciprocity occurred and its strength agree with time domain simulation and measurements.

Data accessibility. The datasets supporting this article have been uploaded as part of the supplementary material. Proofs of some of the important results are provided in 'manuscript_RSPA_supplementary.pdf' MATLAB code developed to generate the figures are attached as a zipped file. The header of 'StateSpaceModelling_CRLH2.m' in the CRLH Example directory is the main script that generates the results appearing in figures 5–8. The files 'StateSpaceModelling_NLRHTLModified_Nov2020.m' and 'script_evb_bvp_nov2020.m' are used to post process and simulate the NL RH TL example.
Authors' contributions. G.M. designed and fabricated the nonlinear transmission line. S.E. carried out the theoretical derivations and numerical analysis. Both authors did the measurements.
Competing interests. We declare we have no competing interests.
Funding. No funding has been received for this article.

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
