## [Peer Review File · Royal Society Open Science]

Review History

RSOS-210367.R0 (Original submission)

Review form: Reviewer 1

Is the manuscript scientifically sound in its present form?

Yes

Are the interpretations and conclusions justified by the results?

Yes

Is the language acceptable?

Yes

Do you have any ethical concerns with this paper?

No

Have you any concerns about statistical analyses in this paper?

No

Recommendation?

Accept as is

Comments to the Author(s)

This paper is acceptable as a contribution to Royal Society Open Science

Review form: Reviewer 2

Is the manuscript scientifically sound in its present form?

Yes

Are the interpretations and conclusions justified by the results?

Yes

Is the language acceptable?

Yes

Do you have any ethical concerns with this paper?

No

Have you any concerns about statistical analyses in this paper?

No

Recommendation?

Accept as is

Comments to the Author(s)

The index "m" in ω_m to refer to modulation frequency is easily confused with other integer indexes. Maybe this ω_m could be rewritten as $\hat{\omega}$ to avoid any confusion (the same could apply to the associated magnitudes).

Also, for the sake of simplicity, $\tilde{\omega}_k$ could be simply renamed as ω_k (the same for β).

When the horizontal axis in the figures refers to β , it is recommended to always use β/π .

"a.u." is not defined.

Review form: Reviewer 3

Is the manuscript scientifically sound in its present form?

Yes

Are the interpretations and conclusions justified by the results?

Yes

Is the language acceptable?

Yes

Do you have any ethical concerns with this paper?

No

Have you any concerns about statistical analyses in this paper?

No

Recommendation?

Accept with minor revision (please list in comments)

Comments to the Author(s)

This paper develops a circuit approach to analyze the space-time periodic structures, based on the ABCD matrices, they derive the translation operator of the LTP and explore the eigenmodes, dispersion law. The theory adopted in this paper is related to your previous IEEE paper, while two examples are introduced to prove the theory.

The authors describe that the circuit is etched on a substrate and varactors are bonded in place, could the authors supplement the circuit and the experiment figure, and make some supplementary on the experimental measurement?

Decision letter (RSOS-210367.R0)

Dear Dr Elnaggar

On behalf of the Editors, we are pleased to inform you that your Manuscript RSOS-210367 "Properties of Translation Operator and the Solution of the Eigenvalue and Boundary Value Problems of Arbitrary Space-time Periodic Metamaterials" has been accepted for publication in Royal Society Open Science subject to minor revision in accordance with the referees' reports. Please find the referees' comments along with any feedback from the Editors below my signature.

Please submit your revised manuscript and required files (see below) no later than 7 days from today's (ie 07-Apr-2021) date. Note: the ScholarOne system will 'lock' if submission of the revision is attempted 7 or more days after the deadline. If you do not think you will be able to meet this deadline please contact the editorial office immediately.

on behalf of Dr Derek Abbott (Associate Editor) and R. Kerry Rowe (Subject Editor)
 openscience@royalsociety.org

Reviewer comments to Author:

Reviewer: 1

Comments to the Author(s)

This paper is acceptable as a contribution to Royal Society Open Science

Reviewer: 2

Comments to the Author(s)

The index "m" in ω_m to refer to modulation frequency is easily confused with other integer indexes. Maybe this ω_m could be rewritten as $\hat{\omega}$ to avoid any confusion (the same could apply to the associated magnitudes).

Also, for the sake of simplicity, $\tilde{\omega}_k$ could be simply renamed as ω_k (the same for β).

When the horizontal axis in the figures refers to β , it is recommended to always use β/π .

"a.u." is not defined.

Reviewer: 3

Comments to the Author(s)

This paper develops a circuit approach to analyze the space-time periodic structures, based on the ABCD matrices, they derive the translation operator of the LTP and explore the eigenmodes, dispersion law. The theory adopted in this paper is related to your previous IEEE paper, while two examples are introduced to prove the theory.

The authors describe that the circuit is etched on a substrate and varactors are bonded in place, could the authors supplement the circuit and the experiment figure, and make some supplementary on the experimental measurement?

===PREPARING YOUR MANUSCRIPT===

===PREPARING YOUR REVISION IN SCHOLARONE===

- If you are providing image files for potential cover images, please upload these at this step, and inform the editorial office you have done so. You must hold the copyright to any image provided.
- A copy of your point-by-point response to referees and Editors. This will expedite the preparation of your proof.

- Ensure that your data access statement meets the requirements at <https://royalsociety.org/journals/authors/author-guidelines/#data>. You should ensure that you cite the dataset in your reference list. If you have deposited data etc in the Dryad repository, please only include the 'For publication' link at this stage. You should remove the 'For review' link.
- If you are requesting an article processing charge waiver, you must select the relevant waiver option (if requesting a discretionary waiver, the form should have been uploaded at Step 3 'File upload' above).
- If you have uploaded ESM files, please ensure you follow the guidance at <https://royalsociety.org/journals/authors/author-guidelines/#supplementary-material> to include a suitable title and informative caption. An example of appropriate titling and captioning may be found at [https://figshare.com/articles/Table_S2_from_Is_there_a_trade-off_between_peak_performance_and_performance_breadth_across_temperatures_for_aerobic_sc ope_in_teleost_fishes_/3843624](https://figshare.com/articles/Table_S2_from_Is_there_a_trade-off_between_peak_performance_and_performance_breadth_across_temperatures_for_aerobic_scope_in_teleost_fishes_/3843624).

Author's Response to Decision Letter for (RSOS-210367.R0)

See Appendix A.

Decision letter (RSOS-210367.R1)

Dear Dr Elnaggar,

It is a pleasure to accept your manuscript entitled "Properties of Translation Operator and the Solution of the Eigenvalue and Boundary Value Problems of Arbitrary Space-time Periodic Metamaterials" in its current form for publication in Royal Society Open Science.

Please ensure that you send to the editorial office an editable version of your accepted manuscript, and individual files for each figure and table included in your manuscript. You can send these in a zip folder if more convenient. Failure to provide these files may delay the processing of your proof. Specifically, we do not have your data/ESM files.

on behalf of Dr Derek Abbott (Associate Editor) and R. Kerry Rowe (Subject Editor)
openscience@royalsociety.org

Appendix A

Reviewer comments to Author:

Reviewer: 1

Comments to the Author(s)

This paper is acceptable as a contribution to Royal Society Open Science

Reviewer: 2

Comments to the Author(s)

The index "m" in ω_m to refer to modulation frequency is easily confused with other integer indexes. Maybe this ω_m could be rewritten as $\hat{\omega}$ to avoid any confusion (the same could apply to the associated magnitudes).

Also, for the sake of simplicity, $\tilde{\omega}_k$ could be simply renamed as ω_k (the same for β).

When the horizontal axis in the figures refers to β , it is recommended to always use $\beta p/\pi$.

"a.u." is not defined.

We have modified the manuscript as suggested by the reviewer. We have defined a.u. (arbitrary units) as highlighted in the manuscript.

Reviewer: 3

Comments to the Author(s)

This paper develops a circuit approach to analyze the space-time periodic structures, based on the ABCD matrices, they derive the translation operator of the LTP and explore the eigenmodes, dispersion law. The theory adopted in this paper is related to your previous IEEE paper, while two examples are introduced to prove the theory.

The authors describe that the circuit is etched on a substrate and varactors are bonded in place, could the authors supplement the circuit and the experiment figure, and make some supplementary on the experimental measurement?

The supplementary material has been amended with figures and a brief explanation of the hardware and experimental setup.